

# Construction patterns of birds' nests provide insight into nest-building behaviours

Lucia Biddle, Adrian M. Goodman and D. Charles Deeming

School of Life Sciences, Joseph Banks Laboratories, University of Lincoln, Lincoln, United Kingdom

## ABSTRACT

Previous studies have suggested that birds and mammals select materials needed for nest building based on their thermal or structural properties, although the amounts or properties of the materials used have been recorded for only a very small number of species. Some of the behaviours underlying the construction of nests can be indirectly determined by careful deconstruction of the structure and measurement of the biomechanical properties of the materials used. Here we examined this idea in an investigation of Bullfinch (*Pyrrhula pyrrhula*) nests as a model for open-nesting songbird species that construct a "twig" nest, and tested the hypothesis that materials in different parts of nests serve different functions. The quantities of materials present in the nest base, sides and cup were recorded before structural analysis. Structural analysis showed that the base of the outer nests were composed of significantly thicker, stronger and more rigid materials compared to the side walls, which in turn were significantly thicker, stronger and more rigid than materials used in the cup. These results suggest that the placement of particular materials in nests may not be random, but further work is required to determine if the final structure of a nest accurately reflects the construction process.

# INTRODUCTION

Nests, built by most birds, are essential for reproductive success, with considerable amounts of time and energy being spent by some species (*Berg et al., 2006*). Although *Collias & Collias (1984)* and *Hansell (2000)* provide general descriptions of avian nest construction, we know relatively little detail for particular species (*Healy, Morgan & Bailey, 2015*) and few reports quantify the materials used in nests (*Deeming & Mainwaring, 2015*). The choices of the materials within different parts of a nest presumably reflect decisions made by the building bird and appear to have a structural role. Recent studies have tried to determine the factors that affect nest construction both using captive species and by examining nests from the field. *Bailey et al. (2014)* showed that captive Zebra Finches (*Taeniopygia guttata*) select artificial nesting material (string) based on its structural properties and that the experience of the bird influences their choice of materials. Furthermore, birds also show an apparent sensitivity to material length (*Muth & Healy, 2014*). A study on wild Common Blackbirds (*Turdus merula*) showed that the birds appear to select thicker, stronger, more

Corresponding authors
Lucia Biddle, lbiddle@lincoln.ac.uk
D. Charles Deeming,
cdeeming@lincoln.ac.uk

rigid materials for the outer nest wall compared to the cup lining or inner structural wall (*Biddle, Deeming & Goodman, 2015*). There were also significant differences within the cup wall; materials at the base of the cup were thicker, stronger and more rigid than those from the top. This suggests that these birds may have some level of awareness of where and when to place different materials in order to create a nest structure. This has yet to be tested experimentally and the mechanism behind this is unknown but captive Zebra Finches are able to learn to choose between nest materials on the basis of structural properties (*Bailey et al., 2014*; *Muth, Steele & Healy, 2013*).

Other animals have also been shown to select materials in a non-random manner during construction of nests and other structures; beavers (*Castor fiber*) predominantly use branches from willow (*Salix*) and poplars (*Populus*) in the construction of their lodges irrespective of the availability of other species (*Fustec & Cormier, 2007*). Branch materials were also selected on the basis of their thickness and, where alder (*Alnus*) species were used, beavers used thinner branches measuring 1.5–3.5 cm in diameter (*Barnes & Mallik, 1996*). Furthermore, orang-utans have been shown to select materials based on their structural properties: nests were constructed using weaker and more flexible branches for the lining and stronger, more rigid and thicker ones for the main structure (*Van Casteren et al., 2012*).

To date, detailed descriptions of materials used in nest construction are limited to only a few bird species, and these tend to build nests either within cavities (e.g., Blue Tits) or appear to have a more complex internal structure, such as a mud cup within a structural nest wall (e.g., Common Blackbirds). This study furthers our understanding of the variability in nest construction for different passerine species by investigating the structure and functional properties of Eurasian Bullfinch (*Pyrrhula pyrrhula*) nests. These are superficially a cup of woven grasses located within a depression of outer nest material made of twigs (*Bochenski & Oles, 1981*). We hypothesised that the different structural regions of Bullfinch nests would have different physical characteristics and that this would relate to the materials they are composed of.

## METHODS

### Nest characteristics

Thirteen bullfinch nests were collected in September 2014 after the known end of the breeding season. Nests were collected by BTO nest recorders who very carefully removed the nests from their location to ensure that the nests retained their structural integrity and composition. All but one of the nests were collected from the Greater Manchester area in north-west England and the other nest was from Tyne and Wear, north-east England. Although exact construction dates were unknown, dates of clutch initiation were recorded and suggested that construction generally took place around the end of March to the beginning of April. The nests were well packaged in cardboard boxes during transportation to the University of Lincoln in order to reduce the chance of damage.

In Lincoln, each nest was dried before being placed in a plastic bag and stored in a cardboard box at approximately room temperature and humidity. In order to provide a consistent base for comparison each nest was conditioned at 23 °C, 50% relative humidity

in a Sanyo MLR-351H environmental chamber for 7–8 weeks, until they had equilibrated to a constant weight. All testing of materials was performed within 24 h after removal from the cabinet.

The weight of each nest was measured using electronic scales (A & D Company Limited, model FX-3000i) and the depth of nests and their width and length were measured using dial callipers. Overall the nests were elliptical in shape so the diameter of the cup and wall thickness were measured both parallel and perpendicular to the long axis of the cup. The volume of the nest cup was measured by lining the cup with domestic cling film before filling the nest cup level to the top edge with 5 mm diameter solid-glass beads (Sigma–Aldrich), which were weighed to allow calculation of volume based on a pre-determined density (*Biddle, Deeming & Goodman, 2015*).

## Nest deconstruction

Before deconstruction nests were visually examined to identify any distinct regions. The main easily identifiable regions identified were: the outer nest, which consisted of pieces of plant material, typically twigs which were loosely interwoven, and the cup wall, in general constructed of roots and grass culms tightly interwoven into a cup like shape (Fig. 1). For two nests a distinct 'secondary' cup was present between the cup and the outer material; this was included as part of the cup and was not tested separately. Variation in the vertical plane of the outer nest was investigated by separating the nests into a lower 'basal' region, defined as anything below the external base of the cup, and an upper 'top' region, which were effectively the sides of the nest around the cup (Fig. 1A). There was no obvious vertical variation in the cup therefore it was analysed as a whole.

Nests were carefully deconstructed by one person (LE Biddle) into their regions by separating elements using forceps and taking care not to damage the materials. The walls of the outer nest were first removed until the lower limit of the outer part of the cup was reached. The cup was then removed to leave the base of the outer nest (Figs. 1B and 1C).

In order to investigate variation in the morphology of the nest construction materials, six pieces of nest material were selected at random, using a random number generator, from each region across all nests: cup, base, and upper nest (from areas both parallel and perpendicular to the long axis). For each sample the length (mm) and if present the number of lateral (side) branches were recorded. In addition, the degree of taper of the main axis of the sample was investigated by measuring the diameter using callipers at the widest end of the sample (base) and at base +2 cm—the difference between the two values provided a measure of the degree of taper (reduction in sample thickness).

## Mechanical analysis

Mechanical analysis took place to determine if nests built by Bullfinches had a similar arrangement in the structural components to those seen in nests of other species such as Common Blackbirds (*Biddle, Deeming & Goodman, 2015*), where the outer nest materials were shown to be significantly thicker, stronger and more rigid that the cup lining materials. Furthermore, it would allow comparisons with the structural properties of nests from non-avian species.

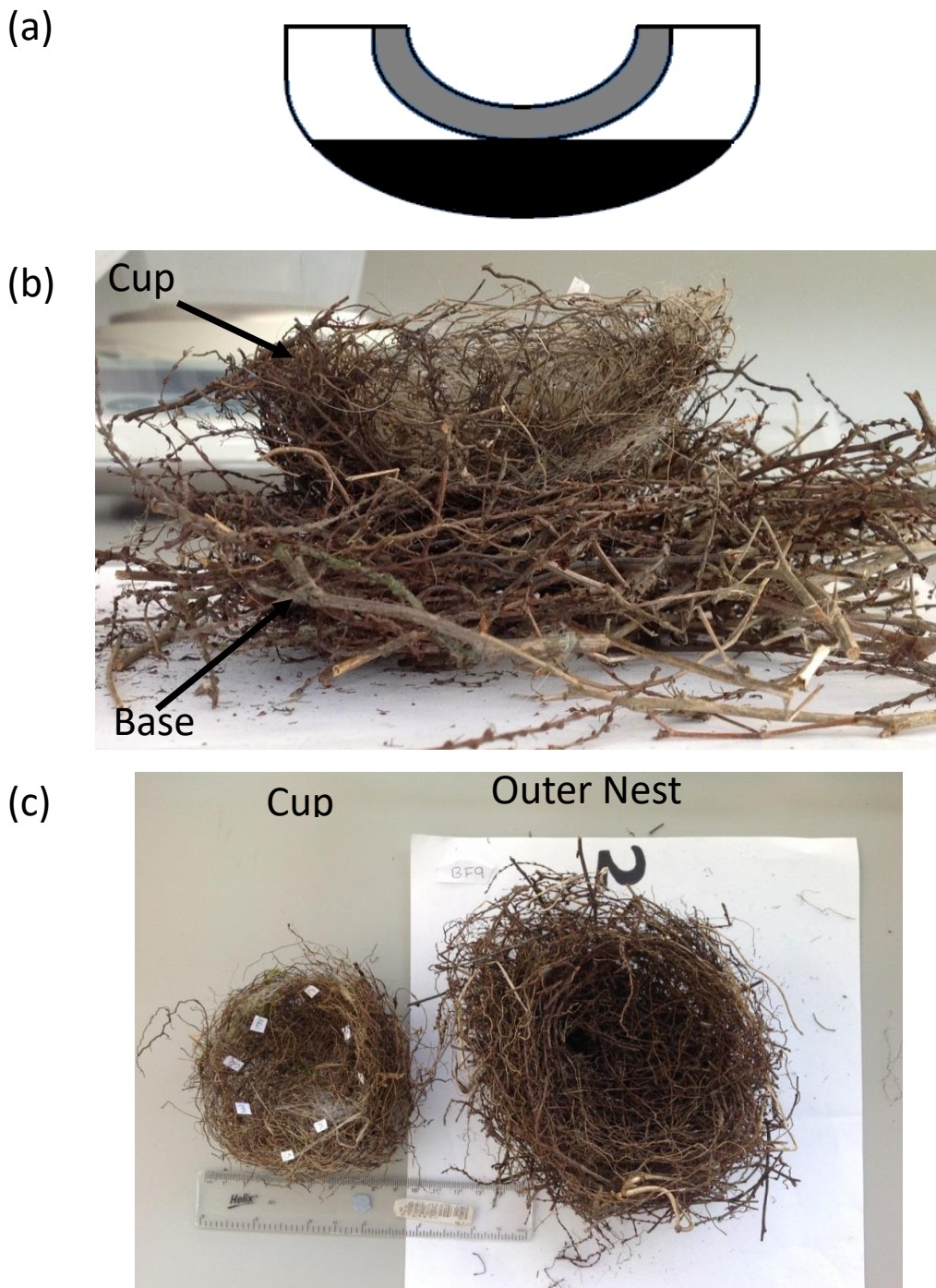

**Figure 1** (A) Bullfinch nest deconstruction regions. Grey, Cup wall; White, Outer nest top; Black, Outer nest base. (B) A Bullfinch nest with the upper outer nest removed to reveal the cup *in situ* and base of the nest. (C) A Bullfinch nest deconstructed into the cup and outer nest components.

After deconstruction, all construction elements from the outer nest were arranged in order of increasing size and the 20 thickest samples were selected from each vertical region for mechanical testing. Only samples with a diameter greater than 0.3 mm were tested; thinner samples were too small to reliably test because of the very low forces generated during bending. For the nest walls, samples were taken from sides both parallel and perpendicular to the main axis of the nest and for the base of the nest samples were taken irrespective of orientation. The same sampling process was applied to materials within the cup, but this was not separated into vertical regions. For the cup, the 20 thickest samples were selected.

All samples were then subjected to three-point bending tests using an Instron universal testing machine model 4443 fitted with a 100 N load cell (*Biddle, Deeming & Goodman, 2015*). Before testing the midpoint of the samples was measured using Mitutoyo, digital callipers (Accuracy of $\pm 0.02$ mm) and the number of hollow samples were recorded. A pushing probe of radius 5 mm was lowered until it just touched the sample placed on two supports on either side. A minimum span-to-depth ratio of 20 was used for each sample in order to limit the effects of shear (*Vincent, 1992*). The crosshead was then automatically lowered at a rate of 10 mm min$^{-1}$ causing the sample to bend until it eventually failed. An interfaced computer produced a graph of force *versus* displacement allowing the structural properties of the beam to be calculated (*Gordon, 1978*).

The bending rigidity, ($EI$ in Nm$^2$; Eq. (1)) of a uniform beam is the resistance of that beam to curvature, where $\frac{dF}{d\delta}$ is the initial slope of the force displacement curve. Bending strength, or maximum bending moment ($M$ in Nm), is given in Eq. (2) where $F_{max}$ is the maximum force (N) a sample will withstand before it fails and $L$ is the distance between the supports in metres.

$$EI = L^3 \left( \frac{dF}{d\delta} \right) / 48 \tag{1}$$

$$M = F_{\text{max}} L / 4 \tag{2}$$

Samples (50, 6.4%) that slipped from their supports during testing were excluded from subsequent analysis.

## Statistical analysis

Paired $t$-tests showed there to be no significant difference in wall thicknesses between the different orientations (parallel and perpendicular) in relation to the long axis of the nest (paired $t$-test: $t_{12} = -0.34$, $p = 0.741$), so the values were pooled thereafter by taking a mean value per nest prior to analysis. Differences between the cup, the outer nest and the lower and upper regions of the nest were investigated using a general linear mixed model (GLMM) in Minitab (version 17) controlling for nest identity as a random factor. For nest composition, each component present within the cup and the outer nest was expressed as a proportion before being arcsine transformed before analysis in order to normalise the data, and then running a stepwise discriminant analysis (*Britt & Deeming, 2011*) to compare the percentage of each material within the cup and outer nest simultaneously using IBM SPSS

Statistics 21. The significance level for discrimination between the parts of the nest was set at an $F$-value of 3.84 (i.e., $P < 0.05$), which is the default for the test. General linear mixed modelling was used to analyse the differences in the structural properties between the nest regions with nest identity as a random factor to control for the fact that samples from the same nest were related (*Biddle, Deeming & Goodman, 2015*).

## RESULTS

### Nest characteristics and composition

Bullfinch nests were composed of an internal cup (internal volume $49.6 \pm 13.1$ cm$^3$) that was physically distinct and easily detachable from the outer nest (Fig. 1C). The cup was asymmetrical with the long axis being approximately 22% longer than the shorter perpendicular axis (Table 1; paired $t$-test: $t_{12} = 6.04$, $p < 0.001$). There was no significant difference between the overall nest length perpendicular and parallel to the long axis of the cup (Table 1; paired $t$-test: $t_{12} = 1.63$, $p = 0.13$). The wall of the nest showed little variation in thickness between the base and upper wall of the nests (Table 1; paired $t$ test: $t_{12} = 0.585$, $p = 0.57$).

The cup wall thickness was not significantly different at any of the four positions measured around the nest (GLMM: $F_{3,48} = 0.24$, $p = 0.868$). However, the outer wall was approximately four times thicker than the wall of the cup and the cup depth was about half of the nest height (Table 1).

The mean total nest mass was 14.7 g with the outer wall mass being significantly heavier than the cup material (Table 1; paired $t$-test: $t_{12} = 5.15$, $p < 0.001$). A significant difference was also seen between the upper and basal regions of the outer wall, with the base being significantly heavier (paired $t$-test: $t_{12} = -7.48$, $p < 0.001$). In the outer nest there was no significant difference in the distribution of materials relative to the long axis (Table 1; paired $t$-test: $t_{12} = -1.017$, $p = 0.329$). The mass of the outer nest showed the greatest variation between nests (coefficient of variation = 46%), compared to the cup (coefficient of variation = 38%).

The type of materials used varied between regions of the nests. The cup was composed of finer, lighter coloured material which was usually grass culms, roots or thin twigs (Fig. 2A), visually this structure seemed more tightly woven than the outer nest. The interior of the cup was not usually lined with animal-derived materials although some hair or fur was found within the material of the cup in around half the nests.

The outer material of the nest was not as tightly packed as the cup, becoming more loosely bound at its extremities. It was mainly composed of eudicotyledonous shoots either from trees (twigs) or from herbaceous species (eudicot herbs); roots and grass culms were also present albeit only in small amounts (Fig. 2A). When nests were composed of large numbers of herbaceous eudicot shoots, the individual elements tended to be shorter and more highly branched than woody twigs, and so gave an appearance of being more tightly woven. Longer woody elements tended to be located towards the base of the nest, and in a few cases these woody elements were placed below the cup and incorporated within the herbaceous material. Relatively few pieces of moss, leaf, artificial material, and bark

**Table 1 Mean (±SD) values for structural dimensions of thirteen Bullfinch nests.** Wall thickness is the average of all 4 sides measured.

| Variable | Mean ± SD | Coefficient of variation (%) |
|---|---|---|
| Nest diameter parallel to long axis (mm) | 131.0 ± 25.6 | 20 |
| Nest diameter perpendicular to long axis (mm) | 118.4 ± 20.7 | 18 |
| Ratio of nest diameters | 1.1 ± 0.2 | 18 |
| Cup diameter parallel to long axis (mm) | 82.2 ± 12.3 | 15 |
| Cup diameter perpendicular to long axis (mm) | 67.3 ± 7.6 | 11 |
| Ratio of cup diameters | 1.2 ± 1.3 | 108 |
| Outer wall thickness (mm) (calculated) | 22.1 ± 9.3 | 42 |
| Cup wall thickness (mm) | 6.2 ± 1.6 | 26 |
| Total upper wall thickness (mm) (calculated) | 24.6 ± 10.7 | 44 |
| Base wall thickness (outer nest and cup wall at base) (mm) (calculated) | 26.4 ± 11.1 | 42 |
| Nest height (mm) | 49.7 ± 11.4 | 23 |
| Maximum cup depth (mm) | 23.3 ± 4.5 | 19 |
| Total nest mass (g) | 14.7 ± 5.0 | 34 |
| Outer wall base mass (g) | 10.3 ± 4.8 | 47 |
| Outer wall top mass (g) | 1.2 ± 0.7 | 58 |
| Outer wall top mass parallel to long axis (g) | 0.5 ± 0.4 | 80 |
| Outer wall top mass perpendicular to long axis (g) | 0.6 ± 0.3 | 50 |
| Total outer wall mass (g) | 11.5 ± 5.3 | 46 |
| Cup wall mass (g) | 3.2 ± 1.2 | 38 |
| Cup volume (cm$^3$) | 49.6 ± 13.1 | 26 |

were found in the nests. A few nests contained leaves and bark as well as human-derived materials such as plastic and thread (Fig. 2A). All nests contained an appreciable amount of dust (Fig. 2A). Discriminant analysis showed that the percentage of roots (Wilk's $\lambda = 0.24$, $F_{1,24} = 76.55$), grass culms ($\lambda = 0.20$, $F_{2,23} = 45.66$) and hair ($\lambda = 1.67$, $F_{3,22} = 36.62$) were significantly ($p < 0.05$) higher in the cup than the outer nest (Fig. 2B).

## Mechanical properties of the construction materials

There were significant regional differences in the mechanical properties of the materials used in the construction of Bullfinch nests. In the outer nest the base region was composed of significantly thicker, stronger and more rigid materials than those from the upper regions of the outer nest (Fig. 3; Table 2). Furthermore, samples from the base of the outer nest were significantly less tapered and had more, longer lateral branches compared to those from the upper region of the outer nest (Tables 2 and 3).

There were significant differences between the materials from the upper part of the outer nest and the cup. Samples from the upper outer nest were significantly thicker, stronger and more rigid than those from the cup (Fig. 3; Table 2). Significantly more tapered elements were in the upper outer nest material compared to the material found within the cup (Tables 2 and 3). Sample length and number of lateral branches were similar to the

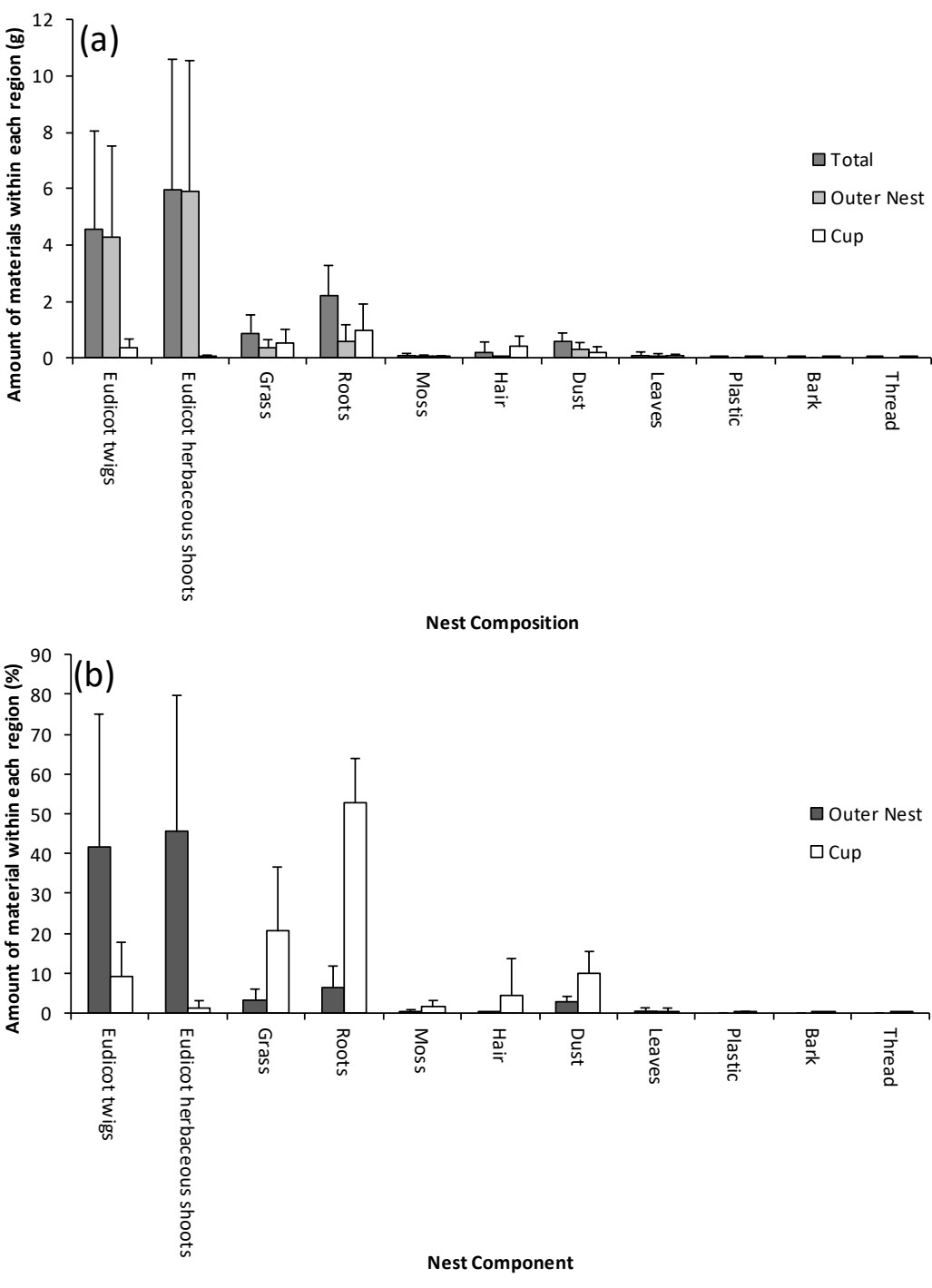

**Figure 2 Within-nest variation in the composition of Bullfinch nests; the nests were separated into the outer nest and cup region along with the overall nest expressed as (A) mass and (B) percentage of the different components.** Values are means +1 standard deviation.

outer nest walls (Tables 2 and 3). There were no significant differences in the percentage of hollow samples between the regions (Tables 2 and 3).

## DISCUSSION

Bullfinches build nests predominately out of twigs and line their nest cups with finer, more pliable plant materials rather than animal-derived materials. Our analysis of the structural properties of the materials showed that the bases of the outer nests were composed of significantly thicker, stronger and more rigid materials than the side walls, which in turn were significantly thicker, stronger and more rigid than materials used in the cup.

### Nest dimensions and construction

Bullfinch nests are predominately constructed by the female bird (*Ferguson-Lees, Castell & Leech, 2011*). The nest dimensions recorded here are comparable to those previously described (*Bochenski & Oles, 1981*) although the total weight of the nest and the cup were slightly lower than previously reported (*Doerbeck, 1963*). The difference in weight may reflect differences in year of collection and/or difference in location, as seen in other passerine species (*Britt & Deeming, 2011*; *Crossman, Rohwer & Martin, 2011*; *Deeming et al., 2012*; *Mainwaring et al., 2012*; *Mainwaring et al., 2014*). The cup was elliptical, as previously reported in Bullfinch (*Bochenski & Oles, 1981*) and Common Blackbird nests (*Biddle, Deeming & Goodman, 2015*), which presumably reflects the shape of the bird's profile as it sits within its nest. Elliptical nest shapes have also been reported in non-avian species; Orang-utans also construct asymmetrical nests where the long axis of the nest is orientated toward the trunk of the supporting tree (*Van Casteren et al., 2012*). It would be interesting to investigate whether birds also orientate their nests *in situ* in line with supporting structures to more effectively distribute the load. Such studies would provide further insight into the selection of nest location in relation to the architecture of supporting structures.

The materials the Bullfinches used during nest construction were similar to those recorded previously (*Nicolai, 1956*; *Newton, 1978*; *Bochenski & Oles, 1981*). However, only two of the nests analysed were lined with hair, which contradicts the report that suggests that all nests are lined with hair (*Ferguson-Lees, Castell & Leech, 2011*). Other reports make no reference to hair (*Nicolai, 1956*) although a nest composed of only hair has been observed (*Doerbeck, 1963*). A significant difference was seen in Bullfinch nests between the materials used in the outer nest and the cup with respect to their insulatory properties (*Hilton et al., 2004*); based on mass there was considerably more hair and dry grass in the cup. Such a distribution suggests that during construction the birds were selecting materials with particular characteristics for different parts of the nest.

### Structural properties of nest materials

The separation of the nest into an outer region and cup is supported by *Bochenski & Oles (1981)* work on Bullfinch nests which showed that the cup was built in a depression within the outer nest material. For the first time we are able to demonstrate that there were significant differences in the structural properties of the various materials used to construct
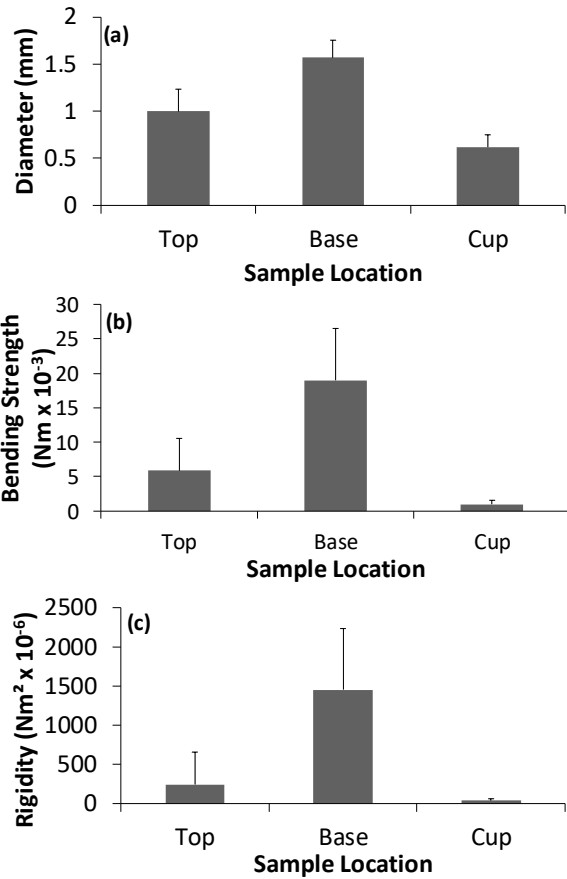

**Figure 3** **Variation in the (A) diameter, (B) strength, and (C) rigidity of the different nest regions for the 13 nests.** Values are means +1 standard deviation.

a Bullfinch nest depending on location within the structure as whole. The positioning of the various materials into different parts of the nest appears to be non-random. The materials in the top of the outer nest were approximately 59% thicker, 474% stronger and 810% more rigid than the materials used in the cup. This may be to provide a supporting framework for the cup and incubating bird whereas the cup is constructed of more flexible materials providing better direct support for the eggs and nestlings.

There were also significant differences in the properties of construction materials in the vertical profile of the outer nest: the material at the base was 59% thicker, 224% stronger and 323% more rigid than the material found in the upper nest surrounding the cup. The non-random distribution of different materials suggests that the birds are therefore able to distinguish between nest construction materials and place the stronger and more rigid ones in the base, which is presumably built first and so supports side walls and the cup from below. Bullfinches are reported to place the cup on top of the outer nest material but not all of the outer nest material is placed below the base of the cup (*Bochenski & Oles, 1981*). This contrasts with the study of Common Blackbird nests which showed no significant vertical difference in the mechanical properties of construction materials in the outer nest materials but did show that at least in the cup lining there were stronger

Biddle et al. (2017), *PeerJ*, DOI 10.7717/peerj.3010

**Table 2** **Results of general linear mixed modelling to test the effect of location within the nest for various structural measures controlling for nest as a random factor in the model.** Materials in the outer nest are compared with those from the upper wall and the base (Fig. 2). The materials from the upper nest wall are also compared to the cup materials (Fig. 2). Values are exponents for changes relative to the characteristics of the upper part of the nest together with accompanying $F$-values, with the degrees of freedom ($df$) shown in the row, and $p$-values in parentheses, also includes the exponent and standard error (SE).

| Variable | Vertical (outer nest: upper vs. base) | | | Wall region (upper outer nest vs. cup) | | |
|---|---|---|---|---|---|---|
| | Change in base relative to upper (exponent ± SE) | Location ($F_{1,12}$) | Nest ($F_{12,12}$) | Change in cup relative to upper (exponent ± SE) | Location ($F_{1,12}$) | Nest ($F_{12,12}$) |
| Diameter | $0.291 \pm 0.026$ | 124.99 (<0.001) | 4.24 (0.009) | $-0.184 \pm 0.028$ | 43.62 (<0.001) | 2.65 (0.052) |
| Bending strength | $0.007 \pm 0.001$ | 52.78 (<0.001) | 2.70 (0.049) | $-0.002 \pm 0.001$ | 14.38 (0.003) | 1.11 (0.431) |
| Rigidity | $555.5 \pm 76.4$ | 52.90 (<0.001) | 4.12 (0.010) | $-153.0 \pm 56.6$ | 7.30 (0.019) | 1.06 (0.460) |
| Length of branches | $5.9 \pm 2.5$ | 5.69 (0.034) | 1.69 (0.188) | $-5.27 \pm 2.47$ | 4.56 (0.054) | 1.59 (0.218) |
| Taper | $-0.044 \pm 0.015$ | 8.82 (0.012) | 1.77 (0.169) | $-0.073 \pm 0.016$ | 20.95 (0.001) | 0.85 (0.605) |
| Number of lateral branches | $0.481 \pm 0.140$ | 11.74 (0.005) | 4.50 (0.007) | $0.168 \pm 0.289$ | 0.34 (0.572) | 0.42 (0.928) |
| Hollow samples (%) | $0.223 \pm 0.275$ | 0.66 (0.433) | 2.22 (0.090) | $-0.327 \pm 0.349$ | 0.88 (0.367) | 0.58 (0.818) |

![PeerJ]

**Table 3  Mean (±SD) values for the physical properties of the samples of nest materials taken from the different parts of the nest.** Sample sizes: length of branches and number of lateral branches (mean *n*—top (11.8), base (6), Cup (6)); Taper (mean *n*—top (11.7), base (6), Cup (6)); percentage of hollow samples (mean *n*—top (16.6), base (20), Cup (18.6)).

| Variable | Top | Base | Cup |
|---|---|---|---|
| | | Position in nest | |
| Construction element length (mm) | 72.48 ± 12.06 | 84.27 ± 16.80 | 61.95 ± 16.23 |
| Taper of construction elements (mm) | 0.20 ± 0.09 | 0.11 ± 0.08 | 0.06 ± 0.06 |
| Number of lateral branches | 1.47 ± 1.02 | 2.44 ± 1.33 | 1.81 ± 1.42 |
| Hollow samples (%) | 20.6 ± 18.5 | 34.6 ± 35.6 | 8.2 ± 7.5 |

and more rigid materials at the base of the cup itself (*Biddle, Deeming & Goodman, 2015*). Captive male Zebra Finches learn to base material choice on structural properties choosing stiffer rather than more flexible string samples to build their nests (*Bailey et al., 2014*). It was also found that during the building process of the nest the stiffer material appeared to be the most effective material compared to more flexible material, with fewer pieces required (*Bailey et al., 2014*). The non-random placement of materials apparently based on their mechanical properties suggests that during nest construction Bullfinches, Common Blackbirds and captive Zebra Finches are able to choose materials. However, laboratory based studies, where birds are provided with limited types of nest materials with relatively similar properties, do not necessarily reflect the natural environment and it would be interesting to investigate what choices Zebra Finches would make if they were provided with more natural nesting materials.

There were also significant differences in the thickness, degree of branching and taper of the construction elements used within Bullfinch nests (Table 2). Again, the distribution of these materials was not random. The base of the nest was composed of thicker, longer and, less tapered elements; these could provide the main supporting members during the early construction of the nest, but also help the base of the nest to support the birds, eggs and rest of the nest structure. Basal elements also exhibited a higher number of lateral (side) branches which may have helped to keep the structure together; it was noted that these side branches were often interwoven, presumably by the birds when building the nests, to the main nest structure. The upper region of the outer nest had significantly more tapered (greater reduction in sample thickness) elements compared to those in the cup. The increased taper may allow the birds to weave the materials together more easily. Whether this pattern is repeated in other species that construct nests primarily out of twigs, e.g., the Hawfinch (*Coccothraustes coccothraustes*; *Ferguson-Lees, Castell & Leech, 2011*) requires further investigation.

Comparison of the mechanical values obtained for Bullfinches with those of Common Blackbirds (*Biddle, Deeming & Goodman, 2015*) shows that the outer nest materials used in the construction of Bullfinch nests were similar in thickness, strength and rigidity. This is despite the fact that Bullfinches are around a third of the mass of Common Blackbirds (*Cramp, 1988*; *Cramp & Perrins, 1994*). These differences may reflect the lack of a mud cup in Bullfinch nests, where the outer nest supports the parent as well as eggs/nestlings.

In Common Blackbirds nests the mud cup is an additional structural element although it is likely to significantly contribute to the nest's self-weight and therefore the nests have thicker support elements to allow for this. There were differences in the composition of the cups between the two species; Common Blackbird nests were lined with dry grass but this was not woven into any structure (*Mainwaring et al., 2014*; *Biddle, Deeming & Goodman, 2015*). By contrast, the elements used in the cup of Bullfinches were generally thinner, weaker and more flexible but were woven to form a discrete structure. These differences may also reflect the lack of the mud cup or the smaller body size of the Bullfinch. The nests were studied ex situ and structural differences may reflect the support provided by the underlying structures—Common Blackbirds often build nests in forks of trees whereas Bullfinches prefer sites, such as a tangle of twigs (*Ferguson-Lees, Castell & Leech, 2011*), which offer less support from below.

A description of nest construction behaviour is unavailable for Bullfinches and so we are unable to categorically state that the birds are selecting the materials that they place in different parts of the nest. However, the differing physical and structural properties of these materials imply that the process of nest construction involves a non-random pattern of behaviour in the placement of various materials as construction of the nest progresses. Given that the amounts of materials used in each region of the nest is typically less than 10 g, we can safely assume that the materials that Bullfinches could use, and do use, in nest construction will be in massive excess in their environment and so their behaviour is unlikely to be limited by material availability. It would be fascinating to test whether captive Bullfinches exhibit any selection of particular nest materials to fit the pattern of usage predicted from the data derived from wild birds.

Further investigations of nest construction in these and other passerine species should investigate whether it is important to relate the size of the clutch and weight of the nestlings and parent to the structural properties of the nest. Moreover, further work is required to investigate the structural properties of nest materials from a greater range of species of different body sizes and nesting locations to help better understand the factors that determine choice of nest material (*Deeming & Mainwaring, 2015*). For instance, do larger, heavier bird species with large clutches build nests with thicker twigs in the base of the nest than smaller species with small clutches?

## Nest construction materials and nest building behaviour

We believe that this study, and that of *Biddle, Deeming & Goodman (2015)*, provides insight into the non-random nature of nest construction behaviour of the species concerned. Our results suggest that Bullfinches might first construct a strong platform to support the nest. They then change the criteria that they use to choose materials for building the side walls of the nest. Finally, the cup is built using a differing arrangement of yet another type of material. Hence, we believe that nest morphology in bullfinches reflects at least three different phases of construction behaviour though we acknowledge that this has not been tested experimentally. We feel that a tri-phasic, but different, pattern of construction is also evident in Common Blackbirds; the first phase of nest construction provides a strong base and surrounding layer prior to the preparation of the mud cup. Finally, this cup is

lined with dry grasses (*Biddle, Deeming & Goodman, 2015*). We appreciate that evidence is indirect but experiments have shown that when captive zebra finches are provided with a choice of nest materials they actively select specific materials for nest building. *Muth, Steele & Healy (2013)* showed that zebra finches preferred a colour for nesting material, but did not prefer a colour for food; this suggests that it is not just colour that determines the behaviour. Another study showed that zebra finches selected the lengths of nest materials to match the size of an entrance hole to a nest box but also changed their behaviour as nest-building proceeds as their experience increased (*Muth & Healy, 2014*). Such studies suggest that during nest construction the birds are deciding upon nest materials of differing properties. It is possible that, if captive birds exhibit such discriminatory behaviour, wild birds could also behave in a similar way in a natural environment. We make no claim about the cognitive abilities of the birds to achieve this process but rather point out that use of different materials in various parts of the nest potentially indicates a change in behaviour as each part is constructed.

However, it could be argued that the finished nest provides little insight into its construction. Birds may bring differing types of materials to the nest in a random manner and only those that are appropriate to that particular location, for instance perhaps stronger elements in the base simply remain in place as nest construction continues; other inappropriate materials are somehow lost. However, if this were the case one would anticipate that, as the nest grew in size and complexity, materials would be retained by the bulk of the structure. In particular thick, strong twigs, more typically associated with the nest base, should be equally distributed throughout the nest and would be found in the cup. It would be predicted that the distribution of materials in the finished structure would be random but we clearly demonstrate that the location of materials is non-random in both Bullfinch and Common Blackbird nests. In addition, *Howell (1943)* describes how the American Robin (*Turdus migratorius*), which is closely related to the Common Blackbird, changes its behaviour as nest construction proceeds by bringing in different materials at progressive stages.

We accept that our interpretation of nest morphology reflecting nest construction behaviour is circumstantial and requires further work. However, our data allow us to postulate a hypothesis that bullfinches are able to distinguish between materials based physical properties and will be selective in their use during nest construction. It would be fascinating to test this hypothesis using captive birds that are offered materials of known physical properties during nest construction. We would predict that the birds would select different materials during the various phases of nest-building. Alternatively, we could apply our methods to bullfinch nests abandoned at different stages of construction, where we would anticipate that a nest abandoned when only the base had been constructed would have similar materials to what we observed in completed nests. Although it was not possible in this study, our results do suggest that a study of the behaviour used by a bird to construct a nest, which is then linked to the subsequent deconstruction of that nest, should be a priority area of further research.

To conclude, our understanding of nest construction behaviour in birds is very poor. To date there are still relatively few detailed descriptions of nest construction behaviour

(*Healy, Morgan & Bailey, 2015*) and despite novel and revealing behavioural studies using captive Zebra Finches building with artificial materials (*Muth, Steele & Healy, 2013*; *Bailey et al., 2014*; *Bailey et al., 2015*; *Muth & Healy, 2014*), relatively little is known about nest construction in the natural environment. We feel that careful deconstruction of nests collected from the wild, when linked with studies into the structural properties of the materials, offers an indirect but valuable insight into nest construction behaviour not only in birds (*Biddle, Deeming & Goodman, 2015*) but also in mammals (*Van Casteren et al., 2012*). Future research is needed to record nest construction behaviour in greater detail whilst quantifying the materials used in a wide range of species with varying nest types. Moreover, there should be greater consideration of the location of nests *in situ*, which will help us develop a broader understanding of factors, e.g., nest site, affecting how nests interact with their natural environment.

## ACKNOWLEDGEMENTS

Many thanks go to Wayne Parry and Richard Winship who supplied the nests used in this study. Many thanks also go to Carl Barrimore and Dave Leech of the *British Trust for Ornithology* Nest Record Scheme for their assistance in publicising our need for bird nests. We are grateful to Tom Pike for statistical advice, and to Fred Rumsey for help with identification of a plant species commonly found in the nests.

### Funding

Studentship was funded by the University of Lincoln. The funders had no role in study design, data collection and analysis, decision to publish, or preparation of the manuscript.

### Grant Disclosures

The following grant information was disclosed by the authors:
University of Lincoln.

### Competing Interests

The authors declare there are no competing interests.

### Author Contributions

- Lucia Biddle conceived and designed the experiments, performed the experiments, analyzed the data, wrote the paper, prepared figures and/or tables, reviewed drafts of the paper.
- Adrian M. Goodman conceived and designed the experiments, analyzed the data, contributed reagents/materials/analysis tools, wrote the paper, reviewed drafts of the paper.
- D. Charles Deeming conceived and designed the experiments, analyzed the data, wrote the paper, reviewed drafts of the paper.

## Data Availability

The raw data has been supplied as a Data S1.

## Supplemental Information

Supplemental information for this article can be found online at http://dx.doi.org/10.7717/peerj.3010#supplemental-information.

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
