# Peer review of "Construction patterns of birds’ nests provide insight into nest-building behaviours"

_PeerJ, doi:10.7717/peerj.3010_

## Round 0.1 · original submission · Major Revisions

As you will see, the reviewers identify a number of issues with your study as it stands. In particular, the conceptual framework could be explained more effectively and tightened up, and more importantly, your interpretation of your results seems to go beyond what the data actually allow. The reviewers also make a number of highly constructive suggestions that I would like you to consider as you revise your paper for further consideration.

Reviewer 1 ·

Basic reporting

The manuscript is generally well written and contains a long list of suitable references that are all relevant to the study. However, my concern about the basic reporting is that the introduction and discussion sections are a bit long winded and generally fail, in my opinion, to focus on the main conceptual issues as highlighted in more depth below.

Major comment

The manuscript, in places, reads like a natural history monograph of bullfinch nests rather than a hypothesis driven study as would be expected in a broad non-taxon-specific journal such as this. Whilst this is not a huge problem in itself, I feel that the manuscript would benefit from being more focused on the non-random use of building materials within nests. Whilst Sue Healy’s lab group have examined some of these issues in captivity, such issues are generally not well understood in wild birds, and so I would suggest that the introduction focuses on this angle far more prominently than it does at present.

Minor comments

P 1, L 1, I am not convinced that the authors have been able to demonstrate any evidence of nest building “behaviours” because they collected nests that had already been built and subsequently bred in, without them witnessing any of the behaviours associated with nest construction. Thus, I think that the title should be changed to better reflect the nature of the study.

P 1, L 16, I’m afraid that I am not convinced that this study is “multidisciplinary” and so I would suggest removing that particular word from the abstract.

P 1, L 18, Replace “nest had particular properties” with “nest served different functions”.

P 2, L 25-30, For me, this section of text is getting to the heart of the matter. Sue Healy and her group have ably shown that nest material use is non-random in the laboratory and the present study does the same thing in the wild, which obviously has some distinct advantages (as well as some disadvantages) over studies of captive birds. Thus, I would suggest revising the introduction to make this point clear.

P 3, L 4, Common blackbird needs a scientific name adding, unless I am mistaken?

P 3, L 19, It may be worth saying that if feathers are not used as nesting materials for their insulatory properties, they may instead be associated with sexual displays or play a role in reducing the abundance of ectoparasites within nests, although I appreciate that the later function has little empirical support.

P 4, L 1-4, Again, this is the sort of study that gets to the heart of the matter. So by saying that there is evidence of non-random placement of nesting materials in mammals, the authors can help to justify their own study by saying that they have studied such issues in birds.

P 4, L 14-15 I’m afraid that I am not convinced by this final prediction because whilst the open and rather flimsy nests of bullfinches may be less well insulated than the nests of hole nesting birds in the absence of attending parents, I would have thought that the embryos within the eggs of both types of bird (open nesting and hole nesting) would have to be kept at very similar temperatures during incubation in order for them to develop normally: see review in: DuRant, S.E., Hopkins, W.A., Walters, J.R. & Hepp, G.R. (2013). Ecological, evolutionary, and conservation implications of incubation temperature-dependent phenotypes in birds. Biological Reviews. 88, 499-509.

Experimental design

The experimental design is sound and the detailed methods are the highlight of the study at present, and will thus hopefully prove useful to those readers wishing to examine similar issues in the future. My only minor comments relate to the reporting of the exact methods followed as they are a bit unclear in places.

Minor comments

P 4, L 19-20, Replace the word “abandoned” because it implies that the nests were abandoned during reproduction, whereas I assume that the authors actually mean that the nests were collected after the end of the breeding season and therefore after the young had fledged the nest successfully?

P 4, L 24: Please provide more details regarding how the nests were removed from the bush or vegetation in which they were built because one suspects that they would have to have been removed rather carefully for them to remain in-tact during the removal process and the subsequent experimental procedures.

P 4, L 25, Reword the “nests were stored dry within individual” text because it does not make sense to me at present.

P 4, L 25: Provide some reassurances that the nests were not damaged during collection or transportation.

P 5, L 7, Replace “nests are typically determined” with “nests have typically been determined”.

P 5, L 12-13, Please clarify why the use of infrared thermography is a better approach to examining these issues than temperature loggers, as it will not be obvious to many readers.

P 5, L 18, I am sorry to be pedantic but in the paragraph above, the authors at least imply that iButtons are not the optimal approach to take, and yet here we find out that the authors have used them in this study!

P 6, L 5-6: Were the pixels positively or negatively related to temperature?

P 6, L 8: Please tell us how many pixels there were so that interested readers can gain some idea of how many pixels were left out of the analyses.

P 6, L 13: Please clarify that these regions were easy to tell apart, if indeed this is true?

P 6, L 24: Please clarify that all nests were deconstructed by the same person, if indeed this was the case?

P 7, L 4: Please tell us what the aim of this mechanical analysis was, as it is unclear at present.

P 7, L 6: Please justify the use of only the 20 thickest samples as not only does it limit the sample sizes somewhat, it also means that the authors never sampled the nesting materials within the nests at random. Thus, please explain and justify your approach to data preparation in more detail here.

P 7, L 27-28: Please tell us how many samples were excluded from the analyses.

P 7, L 33: Please provide accurate P values as simply telling us that the value is larger than 0.05 is not sufficient. This also applies throughout the remainder of the results section.

P 8, L 3: Unless I am mistaken, ANOVA’s do not have random factors and so please explain why you never used more conventional mixed models.

P 8, L 10: So, if your GLM had a random effect, then surely that means that you used a GLMM?

Validity of the findings

The findings of the study are useful and do help to advance our understanding of nest building in birds, although I feel that the focus of the discussion could do with revising following my comments below.

Major comments

If the authors agree with the need to re-focus the manuscript towards the non-random use of building materials within nests, then I feel that the discussion could do some serious revisions to outline exactly how the present study advances our understanding of these issues and of nest composition more broadly.

Minor comments

P 8, L 18, Replace “from an outer” with “from the outer”.

P 9, L 1-2, Without a P value from a statistical test, the authors cannot say that the mass of the outer nests were more variable than the mass of the nest cups.

P 9, L 3-8: None of these statements are backed up by statistical tests either, unless I am mistaken, and so should also not be included in the results section.

P 9, L 9-19: Again, none of these statements are backed up by statistical tests either, unless I am mistaken, and so should also not be included in the results section.

P 10, L 8-10: I would suggest that a significant random effect also indicates that there was a significant difference in the composition of nesting materials between nests.

P 10, L 14-16: Again, I would suggest that a significant random effect also indicates that there was a significant difference in the composition of nesting materials between nests.

P 10, L 28-29: I would suggest rephrasing this text because the presence of foil is surely not a major finding of your study. I accept that it is an interesting methodological result that is useful for other researchers, but hardly a major finding of a study examining nest building birds.

P 11, L 9-10: I agree with this suggestion but please outline what you may expect birds to be doing: for example, would you expect birds’ nests that are oriented with supporting structures to be predated less often than nests that were not oriented with supporting structures?

P 11, L 11-20: This paragraph is largely superfluous in my opinion and could be deleted without too much loss.

P 12, L 1-3: Replace “Future research might be better directed towards” with “Future research could usefully be directed towards”.

P 12, L 10: Please clarify that the nests of common linnets are similar to the nests of bullfinches, if indeed this is the case?

P 12, L 22: I would delete the word “plasticity” here as differences between individual nests that have presumably been built by different females does not reflect plasticity in nest building behaviours.

P 13, L 28: I am not sure that magpies are a suitable species to mention here as their substantial domed nests are presumably built under very different selection pressures to the relatively insubstantial and open cup nests of bullfinches?

P 13, L 30: I would suggest not simply comparing the nests of bullfinches and blackbirds here but instead, broadening out the discussion to birds’ nests more generally.

P 14, L 23: Please briefly explain what “mandibulating” is because it will be unclear to many readers.

P 15, L 16: The reference section contains many sloppy errors and whilst I have highlighted several below, the onus has got to be on the authors to do their best to eliminate such errors.

P 15, L 17: Please insert a space between “2001.” And “Nest characteristics”.

P 15, L 17: Please insert a space between the journal name and the issue number. This issue arises many times throughout the references and so please check them all.

P 16, L 8-9: Please write the journal title in italics.

P 16, L 22: Replace “PerrinsCM” with “Perrins CM”.

P 17, L 29: Please write the Beavers’ scientific name in italics.

P 18, L 11: This DOI number presumably is not correct?

P 19, L 25: Replace “Hansell M” with “Hansell MH”.

P 20, L 1: Replace “used to measure the” with “used to quantify the”.

P 22, L 1: Replace “Regional variation” with “Within-nest variation” as regional variation, to me at least, implies that nests were collected from different regions of Great Britain.

P 23, L 2-3: Please clarify why the sample sizes differ between the different nest parts. And please clarify why you have 16 tops and 20 bases when you only collected 13 nests as it makes little sense to me?

P 23, L 3: Replace “and Cup” with “and Cup”.

P 23, figure 4: I am not sure that “Nest region” is appropriate here either for reasons outlined above.

P 24, table 1: The ‘coefficient of variation’ values are percentage values and yet two are over 100%!

P 25, table 2: Replace “ofvariation” with “of variation”.

P 26, figure 7: I am not sure that “Regional variation” is appropriate here for reasons outlined above.

Additional comments

This study examined various aspects of the composition of bullfinch nests and found good evidence that the birds never used nesting materials at random and instead, for example, incorporated nest materials into the base of the nests that were thicker and stronger than the materials incorporated into the sides of nests. This adds to our understanding of nest building and provides some useful data on bullfinch nests.

Reviewer 2 ·

Basic reporting

Formatting of the references was very poorly done, with many mistakes in spacing. I have attached a PDF with the errors highlighted.

Experimental design

The case was acceptably made for why an investigation into the composition and structure of nesting material is warranted in a previously unreported on species. It does provide new data. However, there is much made throughout the manuscript about the selective abilities of bullfinches choosing material of a particular quality or feature. This is not warranted given the lack of presented plausible alternatives and because the data cannot say anything about the choice or selection of materials by the birds that built the nest. This is not possible without new data on all the possible materials available to the birds while building (to demonstrate their selectivity) or data on their building behaviour (what is in the final structure may not represent what was attempted to be put into the nest). Without additional data much (or all) of the mention on selection, decision-making and awareness of material properties should be removed (and certainly removed from the abstract).

I am also unsure of the purpose of the foil in the study. Is the only influence of the foil to limit convection? Surely the foil will also affect the temperature by reflecting thermal radiation. But beyond this, I’m not sure that the justification for investigating thermal properties in the absence of convection heat loss is sufficiently explained and the research question defined in the introduction.

On Page 6, lines 28-30, there is a mention of six samples selected at random. However, it doesn’t specify how random selection was assured. Physical selection of material would be incredibly open to unconscious bias.

The number of samples that slipped from their supports during mechanical testing is not reported. Also, were the materials replaced?

Validity of the findings

Beyond the lack of data relevant to the selective nature of the bird builders, the validity of findings are OK.

Some minor points: In the results the colour and tightness of the nesting material and weave our reported, but there is no information on how these qualities were assessed.
There are no data or statistical results supporting the correlation among nest dimensions and temperature - which is claimed in the last lines of section 3.2 Thermal characteristics.

The first paragraph of section 4.4 is unconnected to what comes after it and not related to the data presented in the study.

Additional comments

There are several formatting errors (particularly missing spaces) and other minor comments. I have attached a PDF version noting these errors.

Annotated reviews are not available for download in order to protect the identity of reviewers who chose to remain anonymous.

---

## Round 0.2 · Major Revisions

Dear Lucia

Thank you for your revised MS, which has been seen by our two previous reviewers. As you will see, both feel that your paper requires further revision before it is ready for publication, and I agree that in places your interpretation extends beyond what the data can reasonably support. I suspect that, as the reviewers suggest, a shorter less speculative account is what is needed here, and I would urge you to take their comments into account as you revise your paper. For my part, I did find your presentation of your GLM results rather unorthodox -- I'm used to seeing beta-values and confidence intervals, rather than F-ratios, with the random effects as well as the main effects reported. I'm aware that you have taken statistical advice, which suggested you didn't need to report on your random effects, but I'm more of the Andrew Gelman school of thought and feel one should provide all the details of one's analyses so that readers can see exactly what the results showed and assess for themselves. Knowing whether the random effect has any influence on your response variable is valuable information, and can offer further avenues for investigation.

Obviously you may disagree with the suggestion to streamline your paper as the reviewers suggest, and I will obviously understand if you do not wish to shorten your paper and would rather publish elsewhere. I do hope you will consider our reviewers' comments as constructive, and that you will revise your paper and resubmit.

with best wishes,
Louise

Reviewer 1 ·

Basic reporting

The revised version of the manuscript is generally well written and importantly, I do feel that the introduction and discussion sections are now more focused on the main conceptual issues. However, I disagree with the underlying assumption that it is safe to assume that all of the nesting materials used to construct bullfinch nests were freely available in the natural environment as I am unaware of any empirical evidence to support this notion from any bird species. Overall though, many further revisions are needed but I think that the authors have done a decent job in addressing my previous concerns.

Minor comments

P 4, L 1, I would suggest changing the title to “Construction patterns of birds’ nests provide insight into nest-building behaviours” because although the authors have re-arranged the words in the existing title, it remains pretty much the same and is not reflective of the main focus of the study.

P 4, L 2, I suspect that an additional asterisk is needed after Lucia Biddle’s name as there is presently only one by Charles Deeming’s name, and yet both authors are listed as corresponding authors.

P 4, L 12, Replace “based on thermal” with “based on their thermal”.

P 4, L 13, Replace “recorded for a very small” with “recorded for only a very small”.

P 4, L 16, Replace “this idea further in an investigation” with “this idea in an investigation”.

P 4, L 18, Replace “different parts of a nest served different functions” with “different parts of nests serve different functions”.

P 4, L 19, Replace “recording after measuring the insulative” with “recorded after quantifying the insulative”.

P 4, L 22, Replace “results indicate that” with “results indicate that the”.

P 4, L 22, State how the base materials were “different” from materials used in the cup in a bit more details as this is uninformative at present.

P 4, L 23, Replace “materials in the nest is non-random” with “materials in nests is non-random”.

P 5, L 31, Replace “Hansel (2000)” with “Hansell (2000)”.

P 5, L 33, Replace “materials used in a nest” with “materials used in nests”.

P 5, L 34, Replace “made by the bird” with “made by the building bird”.

P 5, L 38-40, The “Pinowski et al …. nest lining” sentence does not really make sense and so please revise it to make it clearer for readers and also to make it fit into the wider text better than at present.

P 5, L 40, Replace “vary with mass” with “vary with the mass”.

P 5, L 43, Replace “which significantly positively” with “which positively”.

P 5, L 48, Replace “play a role as sexual signals” with “play a role in sexual selection”.

P 5, L 56, It is probably best to mention that this study was of wild blackbirds as it may be unclear to at least some readers.

P 6, L 65, Replace “select materials during the construction” with “select materials in a non-random manner during the construction”.

P 6, L 66, Replace “fiber) have been shown to predominantly use branches” with “fiber) predominantly use branches”.

P 6, L 67, Replace “Populus) species in the construction” with “Populus) in the construction”.

P 6, L 74, Replace “study of nest construction” with “study of avian nest construction”.

P 6, L 75, I am not sure that this justification makes sense? Do birds’ nests show greater diversity than mammalian nests? Either way, it’s best to justify this statement in greater detail.

P 6, L 77, Replace “To date detailed descriptions” with “To date, detailed descriptions”.

P 6, L 78, Replace “few bird species, these tend to build” with “few bird species, and these tend to build”.

P 6, L 87, I am not sure this prediction is appropriate because the authors do not actually formally compare the results of this study with the results of their previous study on blackbirds and so I would suggest removing it. At present, you do discuss and compare the results of the two studies but that is normal in discussion sections and I would not describe it as a main prediction.

P 6, L 88, I am also not sure that the use of foil is a good prediction to test because this is a methodological artefact rather than a broad prediction suitable for a broad ecological journal. Thus, it is best to change the emphasis to convection rather than foil.

Experimental design

My only minor comments in this revised version of the manuscript relate to the language used and so the authors should easily be able to deal with these.

Minor comments

P 7, L 99, Replace “where recorded suggesting that” with “where recorded and suggested that”.

P 7, L 101, Replace “during transportation to reduce” with “during transportation to Lincoln University in order to reduce”.

P 7, L 102, Replace “In Lincoln each nest” with “In Lincoln, each nest”.

P 7, L 103, Please clarify that maximum and minimum lengths of time you are referring to here as it is unclear right now.

P 7, L 109, Replace “the depth of the nests and its width and length were measured using” with “the depth of nests and their width and length were measured using”.

P 8, L 138, It is probably best to mention that bullfinches are passerines because this may not be obvious to all readers.

P 8, L 138, Replace “Du Feu” with “du Feu”.

P 8, L 147-149, Two questions arise here: first on which nests were the pilot data performed on?, and second, I thought that all of the nests were tested with foil and so please clarify the situation.

P 9, L 156, Replace “nest temperature was a” with “nest temperature was calculated as a”.

P 9, L 159, Replace “nests were examined” with “nests were visually examined” if indeed this is true?

P 9, L 170, Replace “deconstructed by L. E. Biddle into their” with “deconstructed by one person (L. E. Biddle) into their”.

P 9, L 175, Please clarify if by “samples” you mean pieces of nest material?

P 10, L 183, Replace “took place in order to determine” with “took place to determine”.

P 10, L 183, Replace “if nests created by” with “if nests built by”.

P 10, L 188-195, Please clarify why the thick samples were important and yet the thin samples were unimportant as it is still not clearly justified at present.

P 10, L 199, Tell us how many hollow samples were noted.

P 11, L 222, Replace “nest as a random factor” with “nest identity as a random factor”.

P 11, L 222, Please clarify what you mean by “a percentage” of each component as it is unclear what you are referring to here.

P 11, L 226-227, Please clarify what you mean by “The significance …. of 3.84” as it is unclear what you are referring to here as well.

P 11, L 230, Replace “with nest code as a random” with “with nest identity as a random”.

P 11, L 231, I was not aware that the nest materials were repeatedly tested, which makes this approach unviable?

Validity of the findings

The findings of the study are useful and do help to advance our understanding of nest building in birds and importantly, the revised version of the manuscript is much improved. However, the issue of reporting anecdotal results still remains and pertinently, there are huge chunks of text in the results section that are speculative in nature and thus belong in the discussion.

Minor comments

P 11, L 238, Replace “Surprisingly, there was no” with “There was no”.

P 12, L 243-245, This statement is not backed up by a statistical test, unless I am mistaken, and so should not be included in the results section.

P 12, L 253-257, Again, none of these statements are backed up by statistical tests either, unless I am mistaken, and so should also not be included in the results section.

P 12, L 258-268, Again, none of these statements in the first 11 lines of this paragraph are backed up by statistical tests either, unless I am mistaken, and so should also not be included in the results section.

P 13, L 276, Again, this statement is not backed up by a statistical test, unless I am mistaken, and so should also not be included in the results section.

P 13, L 278, Replace “nest temperature by 0.4°C” with “nest temperatures by 0.4°C”.

P 13, L 297, Replace “seen in the upper” by in the upper“”.

P 13, L 298, Replace “effect of nest (Tables” with “effect of nest identity (Tables”.

P 14, L 308, Please clarify how the materials in the side walls were different to the materials in the nest cups as it may well be unclear to some readers.

P 14, L 311-324, This whole passage of text is not relevant to this study because it just tells us what a bullfinch nest looks like and so please instead clarify how your study advances the field of nest building research.

P 14, L 328, Replace “were lined with hair” with “are lined with hair”.

P 15, L 337, The authors continue to discuss the use of foil here, rather than talking about convection, which is extremely frustrating because the main findings of the study are related to convection and not the foil itself, which is simply a methodological artefact.

P 15, L 346, It is probably best to make it clear that the study by Deeming and Biddle was conducted on British birds which may well be under very different selection pressures to Australian birds.

P 15, L 353, Replace “insight into thermal properties” with “insight into the thermal properties”.

P 16, L 371, Replace “cup and bird whereas” with “cup and incubating bird whereas”.

P 16, L 372, Replace “eggs and chicks” with “eggs and nestlings”.

P 16, L 380, Replace “contrasts to the study” with “contrasts with the study”.

P 16, L 383-384, Earlier the authors say Biddle et al and yet here list all three authors and so please use the correct format both here and throughout the remainder of the manuscript.

P 16, L 384, Replace “Zebra Finches can learn to” with “Zebra Finches learn to”.

P 16, L 389, Replace “and Zebra Finches” with “and captive Zebra Finches”.

P 17, L 393, Replace “more naturalistic nesting materials” with “more natural nesting materials”.

P 17, L 395, Replace “within the Bullfinch nests” with “within Bullfinch nests”.

P 17, L 395, Replace “Again distribution of these” with “Again, the distribution of these”.

P 17, L 399, Please clarify what “lateral” branches are because it is unclear to me at least.

P 17, L 405, It is probably best to briefly remind the reader what “taper” is because it will have slipped out of some readers minds by this point.

P 17, L 409, This study is not a formal comparison with your previous blackbird study and so please amend the text as appropriate.

P 17, L 409 & 414, The authors say “Blackbirds” and “Common Blackbirds” within the space of just a few lines here and so please spell that particular species correctly and consistently both here and throughout the manuscript.

P 18, L 424, Replace “Bullfinches may prefer sites” with “Bullfinches prefer sites”.

P 18, L 433, I disagree here because this argument does not exclude the possibility that some materials may be limited. Have the authors got any references to back up their assertion as they may well be useful here?

P 18, L 435, An alternative idea to this suggestion is that you could provide nesting materials to wild birds which should be easy enough provided you find the nests early enough. Pertinently, these studies may be best performed on species using twigs as nesting material as per the Bullfinches.

P 18, L 438-441, This passage of text reads well but please tell us what that such further work would tell us and how it would advance the field of nest building research.

P 18, L 452-453, This passage of text reads very nicely indeed.

P 19, L 454, Replace “in this study our results” with “in this study, our results”.

P 19, L 455, Whilst I do not disagree with this text, I would argue that it is very difficult to examine nest construction behaviour without deconstructing them!

P 19, L 465-466, Please clarify how those studies would advance our understanding of the function of nests “within their natural environment” because it is presently unclear.

P 19, L 477, Replace “Society B281” with “Society B 281”.

P 21, L 522, Replace “Du Feu” with “du Feu”.

P 23, L 594-595, The scientific name should be in italics.

P 32, L 654, Rephrase “variation in sample morphology” because it is unclear what sample morphology is at present.

Additional comments

The authors have generally done a good job in revising their manuscript and I do think that this version reads better than the previous version. However, much work is still needed to get the manuscript to an acceptable standard and thus, I have made many suggestions that require further thought and attention as detailed in my specific comments. However, I do reiterate that the paper is improved and will hopefully make a valuable addition to the literature.

Reviewer 2 ·

Basic reporting

There are some areas where the wording is not clear or composed of overly long sentences. There are also some areas of questionable word choice.
E.g. page 2, lines 6-8: It is too early to presume that where materials are at the end of the process reflects decisions (this is also a fundamental problem with the whole manuscript)
page 2, lines 16-17 (starting with "...which significantly....") This still does not make sense.
Page 2, lines 18-22: welcome addition, but overly long sentence
Page 3, lines 3-4: Were the Blackbirds used previously not wild? I thought that they were, which makes this statement less significant.
Page 3, repeatedly: Use the term "selected" or "select", when at this stage "used" would be more accurate.
Page 3, line 10: Beaver is still capitalised here (but not earlier)
Page 3, line 19 and 22-23: What is meant by "complicated"? And are the cups described for bullfinches not "complicated"?
Page 3, lines 27-30: The hypothesis on foil comes entirely out of nowhere.
Page 6, lines 27-31: This type of justification should be in the introduction.
Page 7, lines 21-22: The alignment for the Equation numbers is off

Experimental design

I still do not see how the use of foil is relevant, suitable, or able to do what the authors claim (see "Validity of the Findings"). As mentioned previously, foil would do more than just limit convection. Also, the reasoning for including it is under-developed in the introduction. I do not think anything can be drawn from its inclusion.

I am also still unsure about the selection criteria for the material that is subjected to tests. What percentage of total material does 6 and 20 samples represent? Are there trade-offs between amount of material incorporated and thickness? How reflective of the mean are the 20 thickest samples? How was "structurally important samples" determined? By way of example, couldn't lots of thin sticks have the same overall strength as a few large sticks?

Minor issues:
Page 6, lines 2-7: How were these distinct cups reliably recognised? Were there objective criteria used? If something like "contained thinner material" is involved, then doesn't that prejudice your later results?

Validity of the findings

I would have to disagree with the authors response relating to the lack of quantification of some of the results. While they are correct that not everything has to be supported by a statistical test, that would only correctly apply to qualitative descriptive statements. Most, if not all, of the statements by the authors are making comparisons (e.g. "The cup was... more tightly woven than the outer nest."). This is meaningless without comment on how you determined this, and whether the comparison is objective, verifiable, and reliable. On page 9, lines 1-3, there is a distinction between having hair and being "clearly lined with fur/hair" How is this distinguished? Could someone else replicate this determination? All of these statements need to be supported by methodology and, where comparisons are made, supported by quantification and statistical verification; or they should be removed.

The main issue with the whole manuscript is that the final structure does not necessarily give an indication of how it was built. The data do not demonstrate (clearly or otherwise) that the *placement* of particular materials in different parts of the nest is non-random. All it shows is that where they end up (after construction and use) is different across three regions (assuming the items tested are representative - see above). I am aware that the nest were built by birds with a surplus of available material, that is not at issue. Having watched birds building nest, I have witnessed individuals bringing material to the site that is not appropriate for the stage of building, however, it doesn't appear in the final structure because it falls out after the bird attempts to place it. This happens relatively infrequently (so I am not saying choice does not occur), but the final make-up does not indicate any choice or non-random placement. For example, on page 11, lines12-14, the claim is made that hair or dry grass being found in the cup indicates selection of material. If they had brought hair at the start of construction, would you still see it in the final structure? The answer is "No", because it would have fallen out. So the selectivity of material in the final structure in this case is not determined by the bird building the nest, it is determined by other physical properties. This is not the only example - there are a lot of ways that the different materials could become sorted during construction and/or during use that would have nothing to do with how the bird brought material to the nest. Again, I am not saying that non-random placement doesn't happen, only that without additional data directly relating to it there is very little that can be claimed from the existing data.

On the matter of the foil, the authors claim on page 11, lines 23-25 that "convection currents through the nest wall could be important in how nests function as insulating structures." Even ignoring any potential problems with the use of foil, I don't see how their data allows this to be inferred. In order to even begin to make this inference you would need some comparison. Either multiple different nest types that vary in openness of weave, or how the convection varied across temperatures, or something of that nature. Simply finding a different in temperature at a fixed environmental temperature doesn't say much. That whole paragraph is a bit repetitive and does not contribute much. Again, this is even assuming that the use of foil accurately reflects convection heat loss.

The data are only able to support claims about the structural composition of bullfinch nests, so I think that a much shorter less speculative version of the manuscript would be more suited. However, greater effort needs to be made to explain why an individual description of bullfinch nests is needed after the description provided from blackbirds.

Specific issues:
Page 13, lines 4-5: Beyond the zebra finches, you can't state that those species are able to choose materials.
Page13, line 17: The presumption is premature
Page 13, lines 27-28: Again there may be non-choice related reasons for why this is. Something about those characteristics may make them more likely to still be in the nest after construction and use (for example, maybe that thickness, strength or rigidity resists breaking so a similar quantity are needed for structural support, or they are best suited for spanning gaps, or any number of other possibilities).
Page 14, lines 8-18: Largely misses the point. Material shortage, which if it did occur, could lead to the patterns seen. However, it is only one of the possible other explanations (and the easiest straw man to disassemble) that doesn't require selectivity on the part of the birds.
Page 14, line 28: it provides insight into the final product, not the nest construction behaviour.

Additional comments

I think as a general description of the make-up of the nest of bullfinches the data (as they stand) is acceptable. Anything beyond that is beyond the scope of the data and should be removed.

---

## Round 0.3 · Minor Revisions

Dear Lucia,

Thank you for your revision, and I’m happy to say that I think you’ve done a great job with this revision. Having said that, I think there is one (potentially) contentious point remaining, which is the issue of whether the non-random structure of the nest must entail non-random selection and placement of building materials. I understand and agree with the reviewer’s point on this score, although I also appreciate your point that the final product must say something about the process of construction. The problem, as I see it, is that unless you have some means of ‘ground truthing’ your analyses – i.e., testing whether your inferences about the construction process are borne out empirically – then you cannot conclude with full confidence that the structure of a nest accurately reflects the construction process (which is what the reviewer is saying, basically). Although I know that you disagree and do not wish to concede on this point, it does seems to me that, without ground truthing, your results provide preliminary evidence for a hypothesis that remains to be tested. My suggestion therefore is that, if you could include this as a discussion point in your paper, offering the alternative perspectives and the need for further work to test between them, then you would deal with the reviewers’ point in a productive way, and not make inferences that aren’t fully warranted. This would also require you to reword the abstract and other parts of the MS accordingly. I would be happy to accept the paper if it was written along those lines, but I’m hesitant to accept the paper as it stands because I think the interpretation goes beyond what the data reasonably allow.

Consequently, I am happy to provisionally accept your paper, on the condition that you make the suggested changes. If you do not wish to do so, then I’m afraid I will not be able to accept your paper, and I would encourage you to seek publication elsewhere.

I look forward to seeing a new version soon.

With best wishes,

---

## Round 0.4 · accepted · Accept

Many thanks for making the suggested changes!